# Molecular and Clinical Heterogeneity in Hungarian Patients with Treacher Collins Syndrome—Identification of Two Novel Mutations by Next-Generation Sequencing

**DOI:** 10.3390/ijms252111400

**Published:** 2024-10-23

**Authors:** Gréta Antal, Anna Zsigmond, Ágnes Till, András Szabó, Anita Maász, Judit Bene, Kinga Hadzsiev

**Affiliations:** 1Department of Dentistry, Oral and Maxillofacial Surgery, Clinical Center, Medical School, University of Pécs, 7623 Pécs, Hungary; antal.greta@pte.hu; 2Department of Medical Genetics, Clinical Center, Medical School, University of Pécs, 7624 Pécs, Hungary; zsigmond.anna@pte.hu (A.Z.); till.agnes@pte.hu (Á.T.); szabo.andras@pte.hu (A.S.); maasz.anita@pte.hu (A.M.); hadzsiev.kinga@pte.hu (K.H.)

**Keywords:** Treacher Collins syndrome, *TCOF1*, *POLR1D*, incomplete penetrance, NGS, WES

## Abstract

Treacher Collins syndrome (TCS) is a rare congenital craniofacial disorder with variable penetrance and high genetic and phenotypic heterogeneity. It is caused by pathogenic variants in the *TCOF1*, *POLR1D*, *POLR1C,* and *POLR1B* genes, and its major characteristic features are malar and mandibular hypoplasia, downward slanting of the palpebral fissures, and conductive hearing loss. In this study, five patients (two males and three females, age range from 2 to 29 years) with TCS were tested by Next-Generation Sequencing (NGS)-based sequencing and clinically characterized. Genetic analyses detected two deletions and one insertion in the *TCOF1* gene and one missense variant in the *POLR1D* gene. Two novel mutations, c.1371_1372insT (p.Lys458*) in the *TCOF1* gene and c.295 G>C (p.Gly99Arg) in the *POLR1D* gene, were identified. Moreover, two already known mutations, c.4369_4373del (p.Lys1457Glufs*12) and c.2103_2106del (p.Ser701Argfs*9) in the *TCOF1* gene, were detected. The novel *TCOF1* c.1371_1372insT mutation was associated with mild craniofacial manifestations and very rare symptoms of TCS, i.e., developmental delay and moderate intellectual disability. Although incomplete penetrance is a known phenomenon in TCS, surprisingly, the majority of our patients inherited the disease-causing variants from an asymptomatic mother. The unique feature of our study is the observation of causative mutation transmission between asymptomatic family members. Our results expanded the clinical and mutational spectrum of TCS and further confirmed the inter- and intra-familial variability of this disorder.

## 1. Introduction

Treacher Collins syndrome (TCS, OMIM 154500) is a rare, hereditary disorder of craniofacial development belonging to the heterogeneous group of mandibulofacial dysostosis. The disorder is caused by abnormalities in the development of the first and second branchial arches between the 5th and 8th weeks of embryonic development, resulting in various craniofacial malformations [1]. The estimated incidence of TCS is about 1 in 50,000 live births, with no gender or race predilection [2]. Nearly 40% of patients have a positive family history, while in 60% of cases, a de novo mutation is responsible for the development of the disorder [3].

The characteristic hallmarks of the syndrome are malar and mandibular hypoplasia, downward slanting of the palpebral fissures, coloboma of the lower eyelids, sparse eyelashes, retro- and micrognathia, and malformations of the external and middle ears, leading to bilateral conductive hearing loss [4,5]. Minor features such as choanal atresia, cardiac malformations, and developmental delay have also been reported [2,6]. In addition, oral and dental anomalies are often observed, such as a cleft palate, missing or hypoplastic teeth, and crowded teeth [7]. The penetrance of the disorder is considered to be relatively high; however, extreme inter- and intra-familial phenotypic variations can be observed, ranging from medically unrecognizable or mildly affected cases to perinatal death due to severe craniofacial malformations leading to airway obstruction which, in certain cases, makes diagnosis challenging [6,8,9].

TCS is characterized by genetic heterogeneity. To date, four subtypes have been described based on genetic background. The most frequent subtype is Treacher Collins syndrome 1 (TCS1, OMIM 154500) caused by pathogenic variants in the *TCOF1* gene. This subtype shows autosomal dominant (AD) inheritance, and 86% of TCS patients carry a mutation within this gene. The second most frequent form is Treacher Collins syndrome 2 (TCS2, OMIM 613717). Pathogenic variants of the *POLR1D* gene are responsible for this subtype, which is inherited in an AD or autosomal recessive (AR) manner. About 6% of TCS pathogenic variants can be found in the *POLR1D* gene. Treacher Collins syndrome 3 (TCS3, OMIM 248390) follows an AR inheritance pattern. Biallelic variants of the *POLR1C* gene causing TCS3 has occurred in 1.2% of TCS cases. The recently identified *POLR1B* gene is responsible for the novel Treacher Collins syndrome 4 (TCS4, OMIM 618939). This subtype is inherited in an AD manner, and 1.3% of TCS cases belong to this group [10]. TCS is considered a ribosomopathy since the functions of *TCOF1*, *POLR1C*, *POLR1D*, and *POLR1B* are closely linked to ribosome biogenesis [11,12,13]. Causative pathogenic variants have been identified in none of these four genes in about 5–10% of TCS cases. This finding suggests that other genes might be involved in the disease pathogenesis (such as *EFTUD2*) [2].

In this study, we used Next-Generation Sequencing (NGS)-based techniques to discover the genetic defect in a cohort of Hungarian patients with TCS. The aim of our study was the molecular characterization of two novel and two previously reported mutations detected in our TCS patient cohort, along with the clinical characterization of our TCS patients in order to better understand the clinical features and mutational spectrum of TCS.

## 2. Results

### 2.1. Molecular Characterization of the Mutations

The detected mutations in our patient cohort are summarized in Table 1. Mutation screening performed in an international collaboration identified two deletions and one insertion resulting in a premature termination codon in the *TCOF1* gene (NM_001135243) and one missense mutation in the *POLR1D* gene (NM_001374407). 

Among the four mutations detected, two pathogenic mutations were found to be de novo (in Patients 3 and 4), and two mutations (one pathogenic and one variant with unknown significance (VUS)) were inherited from an asymptomatic mother (in Patients 1, 2 and 5).

Sequence analysis of the Comprehensive Hearing Loss and Deafness Next-Generation Sequencing (NGS) gene panel detected a known pathogenic five-base deletion of *TCOF1* c.4369_4373del (p.Lys1457Glufs*12) in Patient 1. In Patient 2, who was the maternal second cousin of Patient 1 and died 15 years ago at the age of two years, targeted Sanger sequencing identified the same mutation in his stored DNA sample [8].

Facial Dysostosis and Related Disorder NGS panel sequencing identified a novel insertion of *TCOF1* c.1371_1372insT (p.Lys458*) in Patient 3 and a novel missense variant, *POLR1D* c.295 G>C (p.Gly99Arg), in Patient 5.

In Patient 4, Whole Exome Sequencing analysis detected a previously described pathogenic deletion, *TCOF1* c.2103_2106del (p.Ser701Argfs*9).

The ACMG classification of the novel insertion of *TCOF1* c.1371_1372insT is pathogenic because this mutation is de novo and predicted to cause disease, loss of function is a known mechanism of TCS, and the identified variant is not found in either gnomAD genomes or gnomAD exomes (PVS1, PM2, PS2). (https://franklin.genoox.com/clinical-db/home, accessed on 22 October 2024). The novel missense variant *POLR1D* c.295 G>C (p.Gly99Arg) is classified as VUS according to ACMG guidelines since this variant is not found in either gnomAD genomes or gnomAD exomes and in silico analyses predict the deleterious effect of this mutation (PM2, PP3).

The 3D modelling of Treacle and POLR1D proteins, which was performed by using the I-TASSER online server, bioinformatic tool, and database [14,15], revealed that both the c.1371_1372insT variant in the *TCOF1* gene and the c.295 G>C variant in the *POLR1D* gene dramatically altered the normal structure of the proteins (Figure 1D and Figure 2D).

### 2.2. Clinical Characterization of Our Patients

The clinical features of our two sporadic and three familial TCS cases are summarized in Table 2. Zygomatic complex hypoplasia, mandibular hypoplasia, downward-slanting palpebral fissures, and bilateral conductive hearing loss, as typical features of TCS, were observed in four out of five probands; however, these features were not so characteristic in Patient 3. Patient 2 displayed typical TCS symptoms as well, but he did not suffer from bilateral conductive hearing loss. Atresia of the external auditory canals was noted in four probands with the *TCOF1* mutation. Two of the four patients with conductive hearing loss (P1, P5) had a BAHA (Bone-Anchored Hearing Aid), and one (P4) had another type of bone conduction hearing device. Microtia was observed in three patients; three patients had a dysplastic ear; and four patients displayed low-set ears. Among the frequent clinical features delayed speech development was noted in two probands, and preauricular hair displacement together with coloboma of the lower lid as well as partial absence of the lower eyelashes was observed in one patient. A cleft palate was observed in one proband, and a high-arched palate was noted in two patients. Regarding dental status, delayed primary tooth eruption, small, widely spaced teeth, and a slight level of plaque were noted in Patient 3. Moreover, this patient displayed developmental delay, moderate intellectual disability, and absence of speech. None of the following features were present in our patient cohort: choanal stenosis/atresia, cardiac malformation, renal malformation, microcephaly, and limb anomaly. In addition, no patients required intubation or tracheostoma during the neonatal period. Thus, a high degree of inter-familial variability in phenotypic manifestation was observed in our patient cohort.

Patient 1 and 2 belong to a large family (Figure 3), which was previously described by our research group [8]. Both male probands suffered from severe craniofacial malformations, and one of them had conductive hearing loss, whereas the mothers of the probands carrying the same mutation in the *TCOF1* gene were completely unaffected. In Patient 5, a similar phenomenon was observed. Although the female proband with the *POLR1D* mutation showed typical features of TCS, the mother of the proband carrying the same mutation was asymptomatic. Thus, extreme intra-familial phenotypic heterogeneity was noticed. It is important to mention that we had a limited amount of clinical data for Patient 2, as he died due to acute aspiration at the age of 2 years.

## 3. Discussion

The diagnostic criteria of TCS were first described by the English ophthalmologist Edward Treacher Collins in 1900 [4]. It is typically a bilateral disease, affecting mainly the eyes, ears, and facial bones with varying severity, whereas intelligence and fertility are normal. TCS is caused by the impaired development of the first and second branchial arches during the 5th to 8th weeks of fetal development. During embryonic development, these branchial arches contain an abundance of cranial neural crest cells (CNCCs), which are migratory multipotent progenitor cells derived from the neuroepithelium, which play a critical role in the formation of a properly structured craniofacial region. Perturbed CNCC formation, proliferation, migration, and differentiation at a specific time window of embryonic development lead to craniofacial dysmorphism [9,13,16]. Several reports have suggested that the inhibition of ribosome biogenesis is a causative factor of TCS [2,11,17]. Four genes, namely *TCOF1, POLR1C, POLR1D,* and *POLR1B*, which are supposed to participate in the pathogenesis of TCS, are closely related to ribosome biogenesis [11,12,13]. *TCOF1* encodes a nucleolar phosphoprotein called Treacle, which is involved in the transcription of ribosomal DNA into pre-rRNA by interacting with an upstream binding factor and RNA polymerase I. Treacle also plays a role in the regulation of post-transcriptional pre-rRNA modifications, and both processes are important stages of ribosome biogenesis [18]. *POLR1D* and *POLR1C* encode subunits of RNA polymerase I and III, which contribute to the synthesis of ribosomal RNA precursors and small RNAs [11].

*TCOF1* is a major causative gene in TCS, and it is located on chromosome 5q32-q33.1, contains 27 exons, and encodes a 152kDa protein. Treacle is a low-complexity, serine/alanine-rich protein with three distinct domains, namely unique N- and C-termini and a central repeat domain. The N-terminal is encoded by six exons and contains nuclear export signal and potential nuclear localization signal (NLS) regions. The central domain encoded by exons 6A-16 consists of 11 repetitive motifs with potential phosphorylation sites. The C-terminus is encoded by exons 17–26 and contains several NLS regions in exons 23, 24, and 25 [18].

More than 200 pathogenic variants in *TCOF1* have been published so far [2,6,9,19,20,21]. The majority of them are deletions that range in size from 1 to 40 nucleotides, mostly causing premature termination codons, leading to a truncated Treacle protein or nonsense-mediated mRNA decay [2,18]. Insertions, duplications, and substitutions also occur frequently [22]. Mutational hotspots have been suggested in exons 10, 15, 16, 23, and 24 since more than half of pathogenic mutations are localized here [23]. In a systematic review by Ulhaq and coworkers, the clinical and mutational data of 338 TCS cases with the *TCOF1* mutation from 43 publications were collected and analyzed. They found that of all the *TCOF1* pathogenic variants, 66% were deletions, followed by substitutions (23%), insertions (6%), duplications (3%), and insertions/deletions (2%). Gross deletions were detected in 4.4% of the patients. Moreover, they identified a real hotspot in exon 24 with a 17.75% frequency. Among the variants within exon 24, the c.4369_4373delAAGAA variant (which was reported previously as c.4135delGAAAA) was observed in 25% of the patients [24]. In their study, Teber and coworkers [6] identified 28 pathogenic mutations with a somewhat different mutational spectrum than previous findings by others. It was surprising that the most common mutation (c.4369_4373del) was not present in their patients, and significantly more (50%) single-base substitutions were found compared to those found in previous reports (19.8%) [22,25]. Splendore and colleagues investigated the parental origin of sporadic mutations. Among ten families with sporadic TCS, the mutations were of maternal origin in three cases, and, peculiarly, all of them were the same single frequent recurrent mutation, the 5 bp deletion (c.4135_4139delGAAAA), which is found in roughly 15% of all diagnosed cases [26].

NGS-based techniques such as NGS gene panel sequencing and WES are efficient tools in the diagnosis of various rare diseases, such as TCS. In our patient group, in accordance with the literature, we identified small deletions and an insertion in the *TCOF1* gene by using the NGS-based approach. Two patients carried the most common deletion (c.4369_4373del), and the segregation analysis revealed that both patients inherited the causative variant from their asymptomatic mothers. A known deletion resulting in a frameshift in exon 12 (c.2103_2106del) was identified in Patient 4, and a novel insertion (c.1371_1372insT) leading to an immediate termination codon in exon 9 was identified in Patient 3 (Figure 4). Although, based on the literature, exons 10, 15, 16, 23, and 24 are assumed to be mutational hotspots in *TCOF1*, only one of the mutations identified in the present work was located in a hotspot, i.e., in exon 24. The 3D modelling of the Treacle protein revealed that the novel c.1371_1372insT variant encodes a protein with a dramatically altered structure. However, this novel variant results in an immediate termination codon. The transcribed mRNA is likely to be degraded via nonsense-mediated decay, leading to a reduced amount of functional protein, so finally, this novel variant exerts its function through haploinsufficiency.

Mutation in *POLR1* genes is less frequent among TCS patients. A study by Ulhaq showed that 11.3% of TCS patients have a causative mutation in *POLR1D*, *POLR1C*, or *POLR1B*. In our study, we identified a novel missense mutation (c.295G>C) in the *POLR1D* gene in one patient. This variant results in a glycine-to-arginine amino acid exchange. The cross-species alignment of the amino acid sequence of POLR1D showed that in position 99, glycine is evolutionarily highly conserved. The 3D modelling of the POLR1D protein revealed that this novel variant dramatically altered the protein structure. Thus, it may have an effect on the activity of RNA polymerase I and III due to the appearance of a positive charge in the protein.

A lot of research has been performed to reveal a phenotype–genotype correlation among TCS patients; however, no correlation has been established to date [2,6,27]. In addition, no association has been found between disease severity and the parental origin of the pathogenic mutation in sporadic or familial cases [26]. Based on previous findings, there is a high degree of inter- and intra-familial variation in the clinical phenotype. Among our patients, in accordance with the literature, inter- and intra-familial phenotypic variability was observed (Table 2). Four of our five patients harbouring the *TCOF1* mutation displayed very frequent clinical features of TCS, such as malar and mandibular hypoplasia and downward-slanting palpebral fissures, and three of these four also had bilateral conductive hearing loss. However, variability was seen in the other clinical symptoms. Among these frequent features, all the patients demonstrated atresia of the external ear canal; moreover, microtia, lower-eyelid coloboma, and speech delay were observed in two patients, and facial asymmetry was seen in one patient. Among the rare features, a cleft palate was observed in one patient. The frequency of these clinical features in patients with TCS based on literature data [23] can be found in Table 2.

Delayed motor development (DD) and intellectual disability (ID) are very rare features of TCS (with an overall frequency of 1.7–10%) [23]; only a few cases have been published so far. A patient carrying a familial c. 911 C>T variant in exon 8 displayed DD and mild ID along with the typical features of TCS, except malar hypoplasia [28]. In a large cohort of patients, Vincent and colleagues identified only one patient (1.8%) with ID [2]. In another study by Teber and coworkers, 3 of 26 TCS patients with the *TCOF1* mutation presented with DD. All these patients showed a severe facial phenotype. One patient harboured a base substitution in the start codon (c.3 G>A), supposed to result in a loss of normal gene function, while another had a 16bp deletion in exon 16 (c.574del), and one carried a splice site mutation (c.2629–3 A>G), which was located in intron 16. The latter is inherited maternally, whereas the remaining two occurred de novo [6]. Our patient carrying a novel mutation in the *TCOF1* gene (c.1371_1372insT) presented DD and moderate ID without the profound craniofacial anomalies characteristic of TCS patients. In addition, speech in our patient did not develop at all, contrary to the other patients with DD or ID, presenting a similar speech delay to that mentioned in previous studies. Moreover, her moderate ID was not assumed to relate to conductive hearing loss. The pathogenesis of these rare features is not clear; no recurrent mutation can be observed among these TCS patients with DD or ID. Moreover, the detected causative mutations were spread over the first 16 exons, although the majority of them were localized in the central domain of the TCOF1 protein. However, it is known from expression studies that Treacle contributes to the development of the central nervous system [29,30].

Although the penetrance of the disorder is considered to be high, the clinical assessment of affected patients showed incomplete penetrance. This is particularly acknowledged for the common *TCOF1* 4369_4373delAAGAA mutation [31]. In addition, the transmission of the disease-causing variant from female to male results in a more severe phenotype, which is also a known phenomenon in this disorder. Two of our patients with severe clinical manifestations carried this most frequent *TCOF1* variant (c.4369_4373del). The male patients, who were maternal second cousins, both inherited the causative variants from their asymptomatic mothers (III/2 and III/7). Within this large family (Figure 3), three patients were phenotypically affected (III/3, IV/3, and IV/8), and two of them (IV/3 and IV/8) were clinically and molecularly investigated. Segregation analysis revealed two additional asymptomatic relatives (II/2 and II/4) carrying the familial mutation. The siblings of Patients 1 and 2 were clinically asymptomatic, and due to their young age, genetic testing was not performed according to regulations. This family clearly presents the above-mentioned phenomena; however, the transmission of the causative mutation occurred between asymptomatic family members as well. This phenomenon is unique to our current knowledge. Moreover, incomplete penetrance was observed in *POLR1D* mutation cases as well.

The limitation of our study is that we applied WES- or NGS-based gene panel sequencing for the molecular characterization of our TCS patients. These techniques are not capable of detecting structural variants located in the possibly regulatory non-coding regions of the *TCOF1* gene. In future studies, whole genome sequencing may detect additional regulatory variants, which may explain why pathogenic mutations are found in three asymptomatic mothers.

## 4. Materials and Methods

### 4.1. Participants

Between 2003 and 2023, six patients with suspected Treacher Collins syndrome were recruited to our genetic counselling unit. During the clinical investigation, 24 clinical features were assessed (details are provided in Table 2 and Section 2.2). NGS-based techniques and targeted Sanger sequencing identified disease-causing genetic variants in five of the six patients. These five patients (two males and three females, age range from 2 to 29 years) were enrolled into this study. The main clinical characteristics of our patient cohort are summarized in Table 2.

This study was approved by the Ethics Committee of the University of Pecs (Protocol 8770-PTE 2021). Written informed consent was obtained from all patients or their legal guardians, and peripheral blood samples were collected. All experiments were performed in accordance with the Helsinki Declaration of 1975 and with the Hungarian legal requirements for genetic examination, research, and biobanking (Hungarian law; XXI/2008).

### 4.2. Genetic Data Analyses

Genomic DNA was extracted from peripheral blood leukocytes using an E.Z.N.A. Blood DNA Maxi extraction kit (OMEGA^®^, Bio-tek, Inc., Norcross, GA, USA). The concentration and purity of the extracted DNAs were measured with the NanoDrop 2000 spectrophotometer (Thermo Fisher Scientific, Waltham, MA, USA).

In Patient 1, Comprehensive Hearing Loss and Deafness gene panel testing with 181 genes (including the *TCOF1*, *POLR1C*, and *POLR1D* genes) was performed by NGS at the GENDIA laboratory. Targeted Sanger sequencing of the *TCOF1* gene was carried out at our laboratory for the DNA sample of Patient 2.

For Patients 3 and 4, an NGS-based panel test of Facial Dysostosis and Related Disorders comprising 27 genes (including the *TCOF1*, *POLR1C*, and *POLR1D* genes) was performed in the Blueprint Genetics laboratory.

For identification of the pathogenic variant in Patient 5, Whole Exome Sequencing was performed in our laboratory. Exomic libraries were prepared by using the Illumina DNA Preparation with Enrichment Kit (Illumina, San Diego, CA, USA), and sequencing was performed on an Illumina NovaSeq 6000 instrument according to the manufacturer’s protocol using paired-end 100 bp reads. The mean sequencing depth of on-target regions was 100.4X–123.5X. Base-called raw sequencing data were transformed into FASTQ format by using Illumina software (v2.20, bcl2fastq). Reads were aligned to the human reference genome (GRCh37:hg19) by using a Burrows–Wheeler Aligner [32]. GATK algorithms (Sentieon Inc., San Jose, CA, USA) were used for duplicate read marking, local realignment around indels, base quality score recalibration, and variant calling. Sequencing depth and coverage were calculated based on the alignments. Sequence data for the *TCOF1*, *POLR1B*, *POLR1C,* and *POLR1D* genes were filtered for further analysis.

For classification and interpretation of genomic data, the guidelines of the American College of Medical Genetics and Genomics (ACMG) [33] were followed. Moreover, databases, such as ClinVar (https://www.ncbi.nlm.nih.gov/clinvar, accessed on 22 October 2024) and The Genome Aggregation Database (gnomAD) (https://gnomad.broadinstitute.org, accessed on 22 October 2024), and genome/exome coverage and in silico prediction tools, such as MutationTaster (https://www.mutationtaster.org, accessed on 22 October 2024), PhastCons, and PhyloP, were used (http://compgen.cshl.edu/phastweb, accessed on 22 October 2024). 

For validation of variants identified by NGS panels and Whole Exome Sequencing, Sanger sequencing was carried out. Moreover, for segregation analyses, targeted parental Sanger sequencing was performed with an ABI 3500 Genetic Analyzer (Applied Biosystems, Foster City, CA, USA).

### 4.3. In Silico Analysis of a Novel Pathogenic TCOF1 Variant (c.1371_1372insT) and a Novel VUS POLR1D Variant (c.295 G>C)

In order to investigate the potential effect of the *TCOF1* and *POLR1D* novel variants on protein structure, 3D modelling was performed using the I-TASSER online server. The *TCOF1* c.1371_1372insT variant resulted in a truncated Treacle protein with a premature termination codon at the 458 amino acid position (Figure 1A). This variant dramatically altered the protein structure (Figure 1D). The *POLR1D* c.295 G>C variant resulted in a glycine-to-arginine amino acid change at position 99. This variant occurs in a highly conserved region of the POLR1D protein (Figure 2B), which has an effect on protein conformation (Figure 2D). It is highly probable that the appearance of a positive charge in the protein has an effect on the activity of RNA polymerase I and III. 

I-TASSER (Iterative Threading ASSEmbly Refinement) is a comprehensive approach for predicting protein structures and for structure-based function annotation. The process starts with identifying structural templates from the PDB (Protein Data Bank) using the LOMETS (Local Meta-Threading Server, version 3) threading approach. Subsequently, reassembly of continuous fragments excised from the PDB templates using replica-exchange Monte Carlo simulations is performed to construct full-length atomic models. If no suitable template is discovered, the complete structure is created using ab initio modelling. Clustering simulated decoys with SPICKER identifies low free-energy states. A second phase of fragment assembly, guided by spatial constraints from both the LOMETS templates and the PDB structures by TM-align, refines the models by reducing steric conflicts and enhancing global topology. REMO generates the final full atomic models by optimizing the hydrogen bonding network. Functional insights are acquired by re-threading the 3D models via the BioLiP (ligand–protein binding) database. The I-TASSER server, which is constantly enhanced for accuracy, allows academic users to submit amino acid sequences and receive high-quality predictions of protein structure and function [15].

## 5. Conclusions

Treacher Collins syndrome is a clinically recognizable, non-progressive disorder; however, molecular diagnosis is of great importance for patients with TCS in the prenatal and postnatal periods. High phenotypic variability and incomplete penetrance are characteristic of this disorder, which makes genetic counselling challenging. Since the disorder affects several organs, TCS patients require multidisciplinary team care. The etiology of TCS is not fully understood yet. In about 10% of cases, no causative variants have been identified in the four genes implicated in the pathogenesis of TCS, so other genes might also be involved. Our findings give further evidence for inter- and intra-familial phenotypic heterogeneity that can be observed in this disorder. Moreover, we demonstrated that the transmission of a disease-causing variant may happen between asymptomatic individuals.

## Figures and Tables

**Figure 1 ijms-25-11400-f001:**
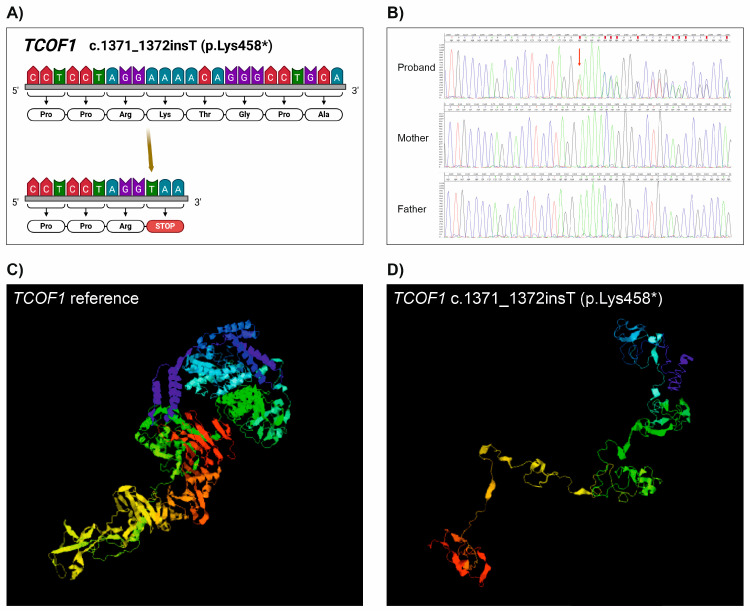
(**A**) Schematic illustration of the identified novel pathogenic *TCOF1* c.1371_1372insT (p.Lys458*) mutation in Patient 3. (**B**) Results of the segregation analysis of this variant by Sanger sequencing. The proband does but her parents do not carry the variants. (**C**) The structure of a normal TCOF1 protein. (**D**) The structure of the altered protein due to c.1371_1372insT DNA insertion. Figure A was created with BioRender.com (accessed on 4 July 2024).

**Figure 2 ijms-25-11400-f002:**
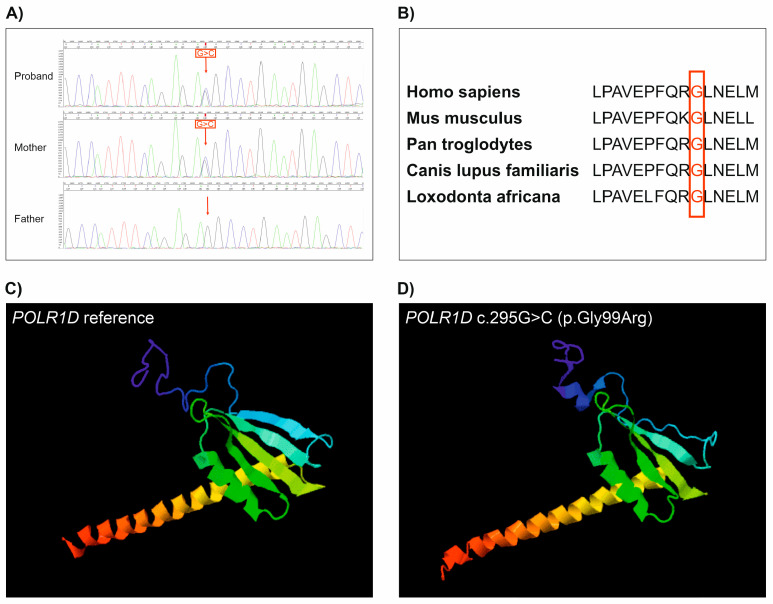
Results of the segregation analysis of the *POLR1D* c.295G>C variant and the 3D structure of the POLR1D protein. (**A**) Sanger sequencing electropherogram of the proband and her parents. The proband and her mother carry the *POLR1D* variant. (**B**) Cross-species alignment of amino acid sequence of POLR1D showed that in position 99, glycine is evolutionarily highly conserved. (**C**) The structure of a normal POLR1D protein. (**D**) The structure of the altered protein due to c.295G>C missense DNA mutation.

**Figure 3 ijms-25-11400-f003:**
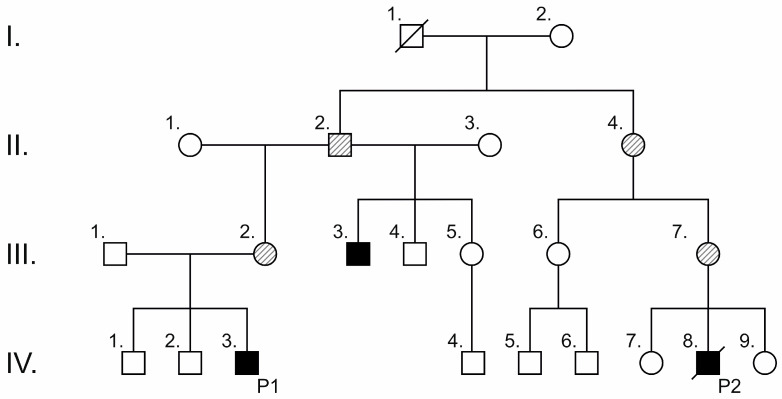
A pedigree of the large family of Patient 1 (P1) and Patient 2 (P2). Squares indicate males, and circles indicate females. Blackened symbols represent affected subjects; clear symbols represent unaffected subjects; and crossed symbols represent deceased family members. Dashed squares and circles indicate asymptomatic males and females, respectively, carrying the causative mutation.

**Figure 4 ijms-25-11400-f004:**
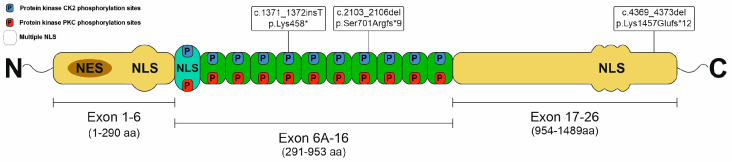
The structure of Treacle protein and localization of *TCOF1* mutations identified in this study.

**Table 1 ijms-25-11400-t001:** Summary of the detected variants in five patients with Treacher Collins syndrome.

	Patients				
	P1	P2	P3	P4	P5
Gene	*TCOF1*	*TCOF1*	*TCOF1*	*TCOF1*	*POLR1D*
TCS disease type	type1	type1	type1	type1	type2
Inheritance	AD	AD	AD	AD	mainly AD or AR
DNA variant	c.4369_4373del ^a^	c.4369_4373del ^a^	c.1371_1372insT	c.2103_2106del	c.295 G>C
Protein	p.Lys1457Glufs*12	p.Lys1457Glufs*12	p.Lys458*	p.Ser701Argfs*9	p.Gly99Arg
Location	Exon 24	Exon 24	Exon 9	Exon 12	Exon 3
Genotype	heterozygous	heterozygous	heterozygous	heterozygous	heterozygous
ACMG classification	Pathogenic (PVS1,PM2,PP5)	Pathogenic (PVS1,PM2,PP5)	Pathogenic (PVS1,PM2,PS2)	Pathogenic (PVS1,PM2,PS2,PP5)	VUS (PM2,PP3)
Transmission	maternal	maternal	de novo	de novo	maternal
Mutation status	known	known	novel	known	novel
Applied method	Comprehensive Hearing Loss NGS panel	Targeted Sanger sequencing	Facial Dysostosis NGS panel	WES	Facial Dysostosis NGS panel

NGS: Next-Generation Sequencing; WES: Whole Exome Sequencing; ACMG: American College of Medical Genetics and Genomics; PVS1: null variant in a gene where loss of function is a known mechanism of disease; PM1: located in a mutational hotspot and/or critical and well-established functional domain without benign variation; PM2: absent from controls in Exome Sequencing Project, 1000 Genomes, or ExAC; PM5: novel missense change at an amino acid residue where a different missense change determined to be pathogenic has been seen before; PS2: de novo in a patient with the disease; PP3: multiple lines of computational evidence support a deleterious effect on the gene or gene product (conservation, evolutionary, splicing impact, etc.); PP5: reputable source recently reports variant as pathogenic but the evidence is not available at the laboratory to perform an independent evaluation. ^a^ The allele frequency of this variant is 1.7 × 10^−6^ according to NFE-gnomAD v4.1.0.

**Table 2 ijms-25-11400-t002:** The main demographic and clinical characteristics of the patients.

Clinical Features			Patients					Occurrence in Patients with TCOF1 Mutation (%) ^b^
			P1	P2	P3	P4	P5	
		Gender	M	M	F	F	F	
		Age at First Examination	After Birth	After Birth	3 mo	2 mo	25 y	
		Age at Time of Study	6 y	2 y ^a^	3.5 y	2.5 y	29 y	
		TCS Type	Type-1	Type-1	Type-1	Type-1	Type-2	
Cranio-facial	Face	Malar hypoplasia/hypoplasia of zygomatic complex	X	X	X	X	X	83–97
		Facial asymmetry	-	N/A	X	-	N/A	52
	Ears	Conductive hearing loss	X	-	X	X	X	83–92
		Atresia of external ear canal	X	X	X	X	-	68–71
		Microtia	X	X	-	N/A	X	10–77
		Dysplastic ear	X	X	-	X	-	N/A
		Low-set ears	X	-	X	X	X	N/A
	Eyes	Downward-slanting palpebral fissures	X	X	X	X	X	89–100
		Coloboma of the lower lid	-	N/A	X	-	-	54–69
		Partial absence of lower eyelashes	-	-	X	-	-	N/A
		Hypertelorism	-	N/A	X	N/A	X	N/A
	Mouth	Cleft palate	-	X	-	-	-	21–33
		High-arched palate	X	-	X	-	-	N/A
		Mandibular hypoplasia/micrognathia	X	X	X	X	X	78–91
		Microstomia	-	N/A	X	X	-	N/A
Central nervous system		Delayed motor development	-	N/A	X	-	-	1.7–10
		Intellectual disability	-	-	X ^#^	-	-	
		Delayed speech development	-	N/A	X *	X	-	57–63
Other		Preauricular hair displacement	-	-	X	-	-	24–49

^a^ Patient 2 died at the age of 2 years (15 years before this study); * speech is absent; ^#^ moderate ID; X means manifestation is present; - means manifestation is not present; N/A means data are not available or not determined; M: male; F: female; ^b^ Marszalek 2021.

## Data Availability

The data presented in this study are available on request from the corresponding author.

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
