# Peer review of "Molecular and Clinical Heterogeneity in Hungarian Patients with Treacher Collins Syndrome—Identification of Two Novel Mutations by Next-Generation Sequencing"

_ijms, 2024, doi:10.3390/ijms252111400_

Round 1

Reviewer 1 Report

Comments and Suggestions for Authors

This study is an excellent short report describing novel mutations in the Treacher Collins syndrome. The genes involved in the TCS are TCOF1, POLR1C and POLR1B.

The authors report 3 structural variants in the TCOF1 gene and one missense variant in the POLR1D gene (four mutations) in 5 patients, and 2 of them are novel variants.

The authors confirmed the inherited variants from an asymptomatic mother and the high penetrance of the syndrome with high intra-familial variability.

Table 1 shows of the summary results of the 4 variants. It would be good if the authors could add the allelic frequency of the 2 deletion variants previously reported in the NFE-gnomAD population observed in P1 and P2.

Table 1 P5 should be revised for maternal transmission and novel variant. Please double check this is correct. 

Table 2 presents the clinical features of the five patients.

Discussion is well-written including details of the protein Treacle, encoded by TCOF1, the main gene in TCS. The majority of the mutations in TCOF1 are deletions leading to truncated Treacle protein and nonsense-medianted mRNA decay.

Although WES can identify most of the causal structural mutations in TCS, the authors should consider mentioning the use of whole genome sequencing since most of the structural variants are located in non-coding regions and additional regulatory variants may explain that the pathogenic mutations are found in 3 asymptomatic mothers 

Author Response

Dear Reviewer,

We appreciate your constructive and valuable comments. Attached please find our responses.

Reviewer 2 Report

Comments and Suggestions for Authors

I would like to thank the authors for their submission and allowing me to review their work.

This is an interesting study on an important topic. However, I would be grateful if you could add further explanations and changes on the following points:

1) TITLE: Page 1, line 4

It is generally better to use full terms and avoid the use of acronyms in the title to ensure clarity.

2) ABSTRACT: Page 1, line 18

I suggest specifying the mean age (± standard deviation), and sex of the study population.

3) INTRODUCTION: Page 2, line 69

I suggest spending some words in describing what NGS is and specifying that it is commonly used in diagnosing genetic causes of congenital and acquired hearing loss, including non-syndromic hearing loss (I suggest citing the following article: Autosomal Dominant Non-Syndromic Hearing Loss (DFNA): A Comprehensive Narrative Review. Biomedicines. 2023;11(6):1616).

4) RESULTS: Page 5, line 139

The study does not provide details on whether the patients utilized bone conduction hearing aids like the BAHA Softband or underwent surgical interventions such as the OSIA system for hearing rehabilitation. Including such information would offer valuable insight into the hearing management strategies used in these cases, especially given the prevalence of conductive hearing loss in TCS patients.

5) Page 11, line 351

I suggest adding a section discussing the limitations of this study.

6) Page 11, line 351

I suggest describing the future prospects and clinical implications of this study, particularly highlighting the new audiological rehabilitation strategies for TCS patients with conductive hearing loss and atresia.

Author Response

(The authors gave the same response as above.)

Reviewer 3 Report

Comments and Suggestions for Authors

Introduction:

- Extend the discussion of Treacher Collins syndrome with additional information on its frequency and effects. cite doi:10.3390/genes12091392.

- Further our understanding of genotype-phenotype correlations in TCS.

- Background with challenges of diagnosing TCS because of its variable expressivity.

- At the end of the introduction, make these study aims more specific.

Methods:

- More details on patient recruitment and inclusion/exclusion criteria [I-3].

- Describe how the sample was selected and explain how this determined your sampling size.

- Please provide details on the NGS and Sanger sequencing protocols with quality control measures.

- Include how variants were classified and interpreted, as well as use of prediction tools.

- Deeper insight into the methodology for 3D protein modeling

Results:

- Table to show clinical traits of patients for comparison

- Give a more in-depth analysis of the novel alleles, commenting on conservation across species and predicted effects.

- If available, show segregation analysis results for all families.

- Consider incorporating figures showing where the identified variants are located in the protein structure.

Discussion:

- Elaborate how these results contrast to previous studies of TCS mutations and the phenotypes observed.

- Describe how these novel variants might be involved in the pathophysiology of TCS.

- Step 2: Discuss the limitations of the study in more depth with potential biases.

- discuss all the possible comorbidities involved in mood and cognitive disorders, as apnea or infections. cite doi:10.3390/children8100921

- Give more concrete pointers on where research is needed next.

Comments on the Quality of English Language

minor errors

Author Response

(The authors gave the same response as above.)
